



# Effects of aerosol-radiation interaction on precipitation during biomass-burning season in East China

**Xin Huang**[1,2,3]**, Aijun Ding**[1,2,3*]**, Lixia Liu**[1,2]**, Qiang Liu**[1,2]**, Ke Ding**[1,2]**, Wei Nie**[1,2,3]**, Zheng Xu**[1,2,3]**, Xuguang Chi**[1,2,3]**, Minghuai Wang**[1,2,3]**, Jianning Sun**[1,2,3]**, Weidong Guo**[1,2,3]**, and Congbin Fu**[1,2,3]

[1]Joint International Research Laboratory of Atmospheric and Earth System Sciences, Nanjing University, 210023, China

[2]Institute for Climate and Global Change Research & School of Atmospheric Sciences, Nanjing University, Nanjing, 210023, China

[3]Collaborative Innovation Center of Climate Change, Jiangsu Province, China

* Correspondence to: Aijun Ding (dingaj@nju.edu.cn)



**Abstract**
Biomass burning is a main source for primary carbonaceous particles in the atmosphere and
acts as a crucial factor that alters Earth's energy budget and balance. It is also an important
factor influencing air quality, regional climate and sustainability in the domain of Pan-
Eurasian Experiment (PEEX). During the exceptionally intense agricultural fire season in
mid-June 2012, accompanied with rapidly deteriorating air quality, a series of meteorological
anomalies was observed, including a large decline in near-surface air temperature, spatial
shifts and changes in precipitation in Jiangsu Province of East China. To explore the
underlying processes that link air pollution to weather modification, we conducted a
numerical study with parallel simulations using the fully coupled meteorology-chemistry
model WRF-Chem with a high-resolution emission inventory for agricultural fires. Evaluation
of the modelling results with available ground-based measurements and satellite retrievals
showed that this model was able to reproduce the magnitude and spatial variations of fire-
induced air pollution. During the biomass-burning event in mid-June 2012, intensive emission
of absorbing aerosols trapped a considerable part of solar radiation in the atmosphere and
reduced incident radiation reaching the surface on a regional scale, followed by lowered
surface sensible and latent heat fluxes. The perturbed energy balance and re-allocation gave
rise to substantial adjustments in vertical temperature stratification, namely surface cooling
and upper-air heating. Furthermore, intimate link between temperature profile and small-scale
processes like turbulent mixing and entrainment led to distinct changes in precipitation. On
one hand, by stabilizing the atmosphere below and reducing the surface flux, black carbon-
laden plumes tended to dissipate daytime cloud and suppress the convective precipitation over
Nanjing. On the other hand, heating aloft increased upper-level convective activity and then
favored convergence carrying in moist air, thereby enhancing the nocturnal precipitation in
the downwind areas of the biomass burning plumes.

**1    Introduction**
Biomass burning, defined as open or quasi-open combustion of non-fossilized vegetative or
organic fuel, is widely used by humans to manage and transform land cover for many
purposes and has been identified as one of the most important disturbance agents in world's
terrestrial ecosystems (Fearnside, 2000). It is a major source of many trace gases and
particulate matters on a regional and a global scale (Andreae and Merlet, 2001; van der Werf



et al., 2006; Ito et al., 2007), contributing significantly to the budgets of trace gases,
greenhouse gases and atmospheric aerosols (Langenfelds et al., 2002). For instance, biomass
burning is estimated to be responsible for almost half of global carbon monoxide (CO)
emission and more than one third of total black carbon (BC) emission (Bergamaschi et al.,
2000; Bond et al., 2013). With tremendous and intensive emission of atmospheric pollutants,
it has been recognized as one of the culprits of regional air pollution (Wiedinmyer et al., 2006;
Ryu et al., 2007) and an important disturber of biogeochemical cycles, especially for those of
carbon and nitrogen (Crutzen and Andreae, 1990; Kuhlbusch, 1998). In the Eurasian
continent, i.e., the domain of Pan-Eurasian Experiment (PEEX) (Kulmala et al., 2015),
biomass burning is a very important source influencing air quality, regional climate change
and sustainability (Chi et al., 2013; Ding et al., 2013ab; Lappalainen et al., 2016). In the East
China, the impact of biomass burning to air quality and regional climate change is particularly
interesting because of the mixing of biomass burning plumes with pollutant from fossil fuel
combustion sources (Ding et al., 2013a; Nie et al., 2015; Xie et al., 2015; Lappalainen et al.,

62    2016).

Biomass burning, including forest fires, savanna fires, peat burning, and crop residue burning
in field, generally features a high emission rate of light-absorbing carbonaceous aerosols
(Reid et al., 1998; Schwarz et al., 2008). The most important one is BC, which is intensively
emitted during biomass burning events due to incomplete combustions (Reid et al., 2005;
Akagi et al., 2011). As the dominant absorber of solar radiation in the atmosphere, BC warms
the Earth-atmospheric system and alters the partitioning of energy between the ground surface
and the atmosphere, thereby modifying atmospheric thermodynamic structures and
modulating hydrological cycles (Krishnan and Ramanathan, 2002; Ramanathan et al., 2005;
Qian et al., 2014; Saide et al., 2015; Ding et al., 2016). These modifications induced by
biomass burning have been detected in many regions, especially for those during forest fires.
Surface temperature decline was extensively observed during forest fires in North America,
Asia and Africa (Robock, 1988, 1991; Procopio et al., 2004; Kolusu et al., 2015). By cooling
the surface and stabilizing the atmosphere, intense forest fire may lead to the inhibition of
cloud formation (Andreae et al., 2004; Koren et al., 2004; Feingold et al., 2005), suppression
in precipitation (Rosenfeld, 1999; Sakaeda et al., 2011), and even temporal shift in onset of
monsoon (Liu et al., 2005; Lau et al., 2006; Zhang et al., 2009). In one word, BC has been
demonstrated to cause a significant perturbation in the radiative energy balance and has even



led to global climate change (Penner et al., 1992; Menon et al., 2002; Ramanathan and
Carmichael, 2008).
Although forest and savanna fires are much less notable in China compared with tropical
America, Africa and Southeast Asia (van der Werf et al., 2006), it is noteworthy that China is
a large agricultural country with the world's top-ranked agricultural production, which is
inevitably accompanied by a tremendous amount of crop residue. Field burning of crop
residue is a common and wide-spread management practice in China during post-harvest
periods for the purpose of clearing farmland and providing short-lived ash fertilization for the
crop rotation (Gao et al., 2002). It is estimated that about 120 Tg crop residues were burned in
field across China every year, far higher than those burned in forest fires and savanna fires
(Yan et al., 2006). Previous studies have documented that field burning of crop residue led to
deterioration in regional air quality during harvest season (Yang et al., 2008; Huang et al.,
2012b; Li et al., 2014). What is worse, this kind of pollution occurs periodically in East China,
particularly during the harvest period of wheat in June (Figure 1). However, studies regarding
its effects on meteorology and climate are still limited. Ding et al. (2013a) reported that
temperature and precipitation were dramatically modified during the harvest season in 2012
according to ground based measurements at a regional background station SORPES in the
Yangtze River Delta region in East China (Ding et al., 2013b). However, there is a lack of a
comprehensive picture of how or through which processes the biomass burning plumes
influenced the air temperature and precipitation and on what scale the aerosol-weather
interactions happened during this case.
Here we conducted numerical simulations for the biomass burning event in East China during
mid-June 2012 based on the online coupled meteorology–chemistry model WRF-Chem (the
Weather Research and Forecasting model coupled with Chemistry) combined with multiple
ground-based measurements and remote-sensing retrievals. The rest of this paper is structured
as follows: Section 2 describes the development of an emission inventory for field burning of
crop residues and how the numerical simulations are configured and designed; in Section 3
we validate the modelling results using available measurements, and then analyse the
perturbations in energy budget and temperature adjustments induced by crop residue burning;
finally, three regions with distinct precipitation changes, located near or downwind from the
burning sites, are selected to discuss in detail. Conclusions are drawn in Section 4.



## 2 Data and Methodology

### 2.1 Emission inventory

Modelling aerosols' radiative effects during this biomass burning event first requires accurate quantification and meticulous characterization of emission from field burning of crop residue. Here, emission intensities of trace gases and particulate matters, specifically including carbon dioxide ($CO_2$), carbon monoxide (CO), methane ($CH_4$), Non-Methane Organic Compounds (NMOCs), nitrogen oxides (NOx), ammonia ($NH_3$), sulfur dioxide ($SO_2$), black carbon (BC), organic carbon (OC), and particulate matter ($PM_{2.5}$ and $PM_{10}$ are particles with aerodynamic diameter less than 2.5 and 10 microns, respectively), were estimated based on a bottom-up method. According to the farming season (available at zzys.agri.gov.cn) and province-level statistics on crop cultivation (NBSC, 2013), we can deduce that intensive agricultural fires in June were mainly related to wheat straw burning as a consequence of the extensively spreading cultivation mode of "winter wheat-summer corn" in East China. Burned biomass at province-level was calculated based on statistical data of crop productions, residue-to-production ratios, percentages of crop residues burned in the field. Emissions of various pollutants were derived from the product of burned mass and experiment results on crop-specific combustion efficiencies and pollutant-sepcific emission factors. The detailed methods and involved datasets are described in our previous work (Huang et al., 2012a). To determine the locations and time of crop residue fires, MODIS (Moderate Resolution Imaging Spectroradiometer) Thermal Anomalies/Fire Daily L3 Global Product (MOD/MYD14A1) combined with burned area product (MCD45A1) were introduced for the purpose of emission spatiotemporal allocations (Giglio et al., 2003; Boschetti et al., 2009). MOD/MYD14A1 provides fire identification by examining the brightness temperature relative to neighbouring pixels. MCD45A1 was also incorporated because its bidirectional reflectance model-based change detection approach has been proved to be capable of presenting a more accurate mapping of smaller fragments of burn scars (Roy and Boschetti, 2009). The spatial pattern of fire detections in Figure 2a indicates that open burning of straw mostly concentrated in northern parts of Anhui and Jiangsu province and got extremely severe on 9 and 13 June, as displayed in Figure 2b. Burning of crop residues dominated local emissions of atmospheric pollutants. Taking BC for instance (Figure 2c and d), emission from field burning of crop residue far outweighed that from industry, power plant, residential activity and transportation combined (Li et al., 2015).



**2.2  Numerical simulation**
The numerical simulations in this study were conducted using WRF-Chem version 3.6.1,
which is an online-coupled chemical transport model considering multiple physical and
chemical processes, including emission and deposition of pollutants, advection and diffusion,
gaseous and aqueous chemical transformation, aerosol chemistry and dynamics (Grell, G. et
al., 2011). The model has been widely utilized to evaluate aerosol-radiation-cloud interactions
and aerosol-boundary layer feedback (Grell, G. et al., 2011; Zhao, C. et al., 2013; Fan et al.,
2015; Huang et al., 2015; Ding et al., 2016). In the present work, we adopted two nested
model domains centred at 115.0 °E, 33.0 °N (Figure 1a). The parent domain with a grid
resolution of 20 km covered the eastern China and its surrounding areas to get synoptic
forcing. The fine resolution of 4 km for the inner one allowed better characterization of small-
scale physical processes, especially those linked to convective motions, cloud formation and
rainfall onset. There were 31 vertical layers from the ground level to the top pressure of 50
hPa, 20 of which were placed below 4 km. The initial and boundary conditions of
meteorological fields were updated from the 6-hour NCEP (National Centres for
Environmental Prediction) global final analysis (FNL) data with a $1° \times 1°$ spatial resolution.
The simulation was conducted for the time period from 20 May to 15 June, 2012, during
which each run covered 60 hours and the last 48-hour modelling results were kept. The
chemical outputs from the preceding run were used as the initial conditions for the next run.
First two weeks were regarded as the model spin-up period, so as to minimize the influences
of initial conditions and allow the model to reach a state of statistical equilibrium under the
applied forcing (Berge et al., 2001; Lo et al., 2008).
Key parameterization options for the WRF-Chem modelling were the Noah land surface
scheme to describe the land-atmosphere interactions (Ek et al., 2003), the YSU boundary
layer scheme (Hong, 2010), and the RRTMG short- and long-wave radiation scheme (Mlawer
et al., 1997). The Lin microphysics scheme that accounts for 6 forms of hydrometer (Lin et al.,
1983) together with the Grell cumulus parameterization was applied to reproduce the cloud
and precipitation processes (Grell, G. A. and Devenyi, 2002) for the coarse domain. Cumulus
parameterization was switched off for the inner domain. Previous studies have shown that,
under highly polluted conditions, the ARI dominated over the aerosol-cloud interaction (ACI)
that is related to aerosols' ability to act as CCN (e.g., Rosenfeld et al., 2008; Fan et al., 2015).
Since the focus of this study is on ARI, the prognosed aerosol was disabled to act as cloud





condensation nuclei (CCN) or ice nuclei (IN) in our simulations and therefore the effects from
ACI were not accounted for. For the numerical representation of atmospheric chemistry, we
used the CBMZ (Carbon-Bond Mechanism version Z) photochemical mechanism combined
with MOSAIC (Model for Simulating Aerosol Interactions and Chemistry) aerosol model
(Zaveri and Peters, 1999; Zaveri et al., 2008). Aerosols were assumed to be spherical particles.
The size distribution was divided into four discrete size bins defined by their lower and upper
dry particle diameters (0.039–0.156, 0.156–0.625, 0.625–2.5, and 2.5–10.0 μm). Aerosols in
each size bin were assumed to be internally mixed and their optical properties, including
extinction coefficient, single-scattering albedo (SSA) and asymmetry factor, were computed
based on Mie theory (Fast et al., 2006) and volume averaged refractive indices (Barnard et al.,
2010). Similar model configurations and settings have achieved good performance in our
previous simulations over the eastern China (Huang et al., 2015; Ding et al., 2016). Detailed
configurations and domain settings are listed in Table 1.
Both natural and anthropogenic emissions were included for the regional WRF-Chem
modelling in the present work. Typical anthropogenic emissions were obtained from the
Multi-resolution Emission Inventory for China (MEIC) database (Li et al., 2015), in which
emissions sources were classified into five main sectors: power plants, residential combustion,
industrial processes, on-road mobile sources, and agricultural activities. This database
covered most of anthropogenic pollutants, such as $SO_2$, NOx, CO, volatile organic
compounds (VOCs), $NH_3$, PM, BC, and OC. VOCs emitted from typical anthropogenic
activities and aforementioned crop residue burning were speciated into model-ready lumped
species using profiles for Carbon-Bond Mechanism (Hsu et al., 2006). The biogenic VOC and
NO emissions were calculated online by using the Model of Emissions of Gases and Aerosols
from Nature (MEGAN) embedded in WRF-Chem (Guenther et al., 2006). More than 20
biogenic species, including isoprene, monoterpenes (e.g., α-pinene and β-pinene) and
sesquiterpenes, were considered and then involved in the photochemistry calculation.
In order to disentangle aerosols' role in radiative transfer and subsequent effects on cloud and
precipitation during this biomass-burning event in the mid-June of 2012, we designed two
parallel numerical experiments. Domain settings and model configurations for these two
simulations were exactly the same as mentioned before, except that one experiment (aerosol-
radiation interaction, ARI) took account of aerosols' perturbations in radiation balance while



the other experiment (CTL) did not include any aerosol's effects on either longwave or
shortwave radiation.
**3    Results and discussions**
**3.1    Fire-induced pollution and observed anomalies in meteorology**
As demonstrated by existing studies (Andreae et al., 1988; Huang et al., 2012b; Ding et al.,
2013a), air quality was dramatically deteriorated and the visibility was impeded during
biomass burning events. We compare the simulated $PM_{10}$ concentration with daily
measurements derived from Air Pollution Index (API) in Figure 3 (If not mentioned specially,
the simulation refers to ARI experiment hereafter). Both observations and simulations
manifested the fact that intensive agricultural fires led to the severe pollution in mid-June.
Since 9 June when the detected fire spots became intense and extensive, $PM_{10}$ concentrations
in northern Anhui and northwest Jiangsu province began to increase, especially for those
regions near the fire location. For instance, the observed daily mean $PM_{10}$ concentrations
reached up to around 250 μg/m$^3$ at Fuyang (FY) and Xuzhou (XZ) and even exceeded 400
μg/m$^3$ at Bengbu (BB) on 9 June (the locations of cities mentioned in this article are labelled
in Figure 2). XZ and BB suffered from the second-round fire smoke two days later, with a
maximum daily mean concentration of 548 μg/m$^3$ observed at BB. Figure 4 illustrates the
satellite-retrieved 660-nm aerosol optical depth (AOD) and SSA from MODIS Aerosol
Product MOD04_L2 (daily level 2 data produced at the spatial resolution of 10 km, collection
6) around 11:00 local time (LT) on 9 June when the first-round of extensive fire pollution
broke out. Their comparisons with modelled spatial distributions of $PM_{2.5}$ and BC column-
integrated mass loadings further confirm model's ability to well reproduce atmospheric
pollution for this event. The AOD observation shows that high aerosol loadings were
concentrating in northeast Anhui and the north-central Jiangsu, shaping a belt of pollution
from the fire sites to the downwind areas. The similar pattern was also simulated by the model.
The $PM_{2.5}$ mass loading was found to exceed 200 mg/m$^2$ near BB, NJ and most parts of
central Jiangsu. This strap-shaped pollution was particularly obvious in terms of BC column
concentrations, which was also consistent with a relatively lower SSA along BB, Yangzhou
(YZ) and Taizhou (TZ).
Along with the severe air pollution and poor visibility, anomalies in meteorology occurred on
9-10, June. Ding et al. (2013a) found that, during these two days, a sharp decline existed in



the observed air temperature in NJ and YZ, compared with weather forecast results and NCEP
FNL data, but the simulations and observations showed a good agreement when the heavy air
pollution was not present before 8 June. At YZ the temperature difference was as high as 5.9
and 9.2 ℃ on 9 and 10 June, respectively. Simultaneously, measured solar radiation intensity
and sensible heat flux showed very low values on 10 June in comparison with non-episode
days. Moreover, local meteorological agency forecasted a convective rainfall to occur in NJ
and surrounding areas in the afternoon of 10 June, with the rainfall centre passing by NJ
around 14:00 LT. However, this forecasted rainfall never happened that day.
On the basis of ground-based measurements, vertical sounding data, remote-sensing images
and their comparisons with numerical simulations, we found that agricultural fires worsen
regional air quality to a large extent and caused a series of anomalies in temperature and
precipitation in the mid-June of 2012. How the biomass burning plumes influenced the air
temperature and precipitation will be the main issue to be addressed in the following
discussions.
**3.2   Perturbations in energy budget and temperature responses**
To better understand aerosols' role in the energy balance on 10 June when precipitation was
evidently modified, radiative forcing in the atmosphere and at the ground surface was
estimated by differentiating the ARI and CTL results (Figure 5). At the surface, daily mean
incident short-wave radiation was weakened by 45.5 W/m$^2$ (averaged over the inner domain)
as the extinction of aerosol was quite large with a satellite-observed 660-nm AOD exceeding
2.0 (Figure 4b). Meanwhile, about 60.4 W/m$^2$ shortwave energy was blocked in the
atmosphere over the inner domain due to the fact that absorbing aerosols were accumulated
on that day (Figure 5b). A positive domain-averaged radiative forcing of +14.9 W/m$^2$ was
simulated at the top of the atmosphere (TOA) on 10 June. Radiation measurements collected
at Heifei (HF) and sensible and latent heat flux recorded at Lishui (in South Nanjing) are
compared with the diurnal variations of corresponding simulations in Figure 6, which
supports that significant radiative perturbations took place at NJ and HF. Substantially
weakened daytime solar irradiance was observed on 10 June, when the peak value of
downwelling shortwave radiation at HF was 618.3 W/m$^2$ at HF and was only 309.7 W/m$^2$ at
NJ. ARI tended to predict lower downwards solar radiation, which was closer to observation
for both cities. Reduction in shortwave energy hitting the surface in turn decreased outgoing





heat fluxes, and therefore simulated sensible and latent heat fluxes at 12:00 LT on 10 June in
ARI decreased by 89.3 and 76.1 W/m$^2$, respectively, compared with CTL experiment.
Spatially, the magnitude of the radiative forcing on 10 June was comparable in northern
Anhui and central Jiangsu, differing from the distribution pattern of fire-induced air pollution
that remarkably concentrated in northern Anhui (Figure 5). As revealed in our previous
estimation, among all components of the ambient aerosols, BC is the most important disturber
of shortwave radiation transfer at the surface and in the atmosphere as well (Huang et al.,
2015; Ding et al., 2016). Although fire emission mostly concentrated in the northern Anhui
and resulted in a high BC concentration of 20 μg/m$^3$ there, high-altitude BC was spread much
more broadly. At an altitude of 2 km, BC concentration around 5 μg/m$^3$ stretched from
northern Anhui to central Jiangsu (Figure 5d). Such distinct distributions between two layers
were partly attributed to the stagnant condition near the surface and stronger horizontal
transport in the upper level. It is emphasized that upper-level BC has higher absorbing
efficiency (Ding et al., 2016). That is why the distributions of both positive radiative forcing
in the atmosphere and negative forcing at the surface generally consisted with BC's spatial
pattern in the upper air.
The perturbations in the energy budget and the following re-allocation gave rise to substantial
modulation in temperature stratification. In comparison with CTL experiment, ARI
experiment predicted an obvious decline in near-surface temperature by considering the
effects of aerosol-radiation interaction. Hourly observed 2-m air temperature was compared
with corresponding simulations by two experiments during the time period from 8 to 15 June.
Model-performance statistics including mean bias (MB), mean error (ME) and root mean
square error (RMSE) are presented in Table 2. As shown, CTL simulation had a systematic
positive bias in 2-m temperature and ARI predicted lower temperature for both areas near fire
locations (BB and XZ) and downwind regions (NJ and SY). The decreases in temperature
were pronounced in BB and XZ with a large difference of approximately 1.2 ℃, which
notably narrowed the gaps with observations. On 10 June when the fire-induced pollution
became intensive, the magnitude of surface cooling was remarkably high near the fire sites
(Figure 5e). For instance, compared to CTL, ARI simulated near-surface temperature at XZ
was cooled by almost 8.0 ℃ at 20:00 LT on 10 June (Figure 7b). In addition to the cooling
tendency of near-surface temperature, aerosols' radiative effects also increased air
temperature at a higher altitude, which were more apparent over the downwind areas (Figure

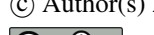



5f). According to the comparisons between simulated temperature profiles by the two parallel
experiments in Figure 7, the warming of air temperature was particularly evident around an
altitude of 2 km at SY with a maximum of 3.0 °C.
The different temperature responses over the source region of fire emission and downwind
areas could be partially interpreted by the fact that near the fire locations pronounced surface
cooling counteracted some of atmospheric warming, which would otherwise elevate upper-air
temperature, through vertical mixing; while for the downwind area where the surface was less
radiatively cooled, the atmosphere was prone to being warmed. As a result of surface cooling
and atmospheric heating, vertical convective motions were weakened, triggering perturbations
in pressure and wind fields (Figure 5e and f). It is obvious that suppressed convection was
generally along with the resultant wind convergence around 2 km and surface divergence,
which may further play a significant role in water vapor transport and then cloud formation.

### 3.3    Effects on cloud and precipitation

In addition to the attenuation of solar radiation and the modulation in temperature gradients,
precipitation also showed many disparities between CTL and ARI simulations. The satellite
observation from Tropical Rainfall Measuring Mission (TRMM) Multisatellite Precipitation
Analysis product (3B42), which provides merged-infrared precipitation information at a
0.25×0.25 ° spatial resolution and has been demonstrated to perform well in East China
(Simpson et al., 1988; Zhao and Yatagai, 2014), was used to evaluate the simulated
precipitation. As demonstrated in Figure 8, ARI experiment agrees better with TRMM
observations in terms of precipitation intensities and also spatial pattern on 10 June.
Specifically, CTL simulation suggested a convective rain in Zone 1 (NJ and its adjacent areas)
around 14:00 LT (the locations of Zone 1-3 are marked in Figure 8c), however the ARI
simulation didn't show any precipitation then, consistent with the TRMM observations.
Besides, ARI displayed enhanced precipitation in northern Jiangsu province. A precipitation
with the intensity of 3 and 5 mm/h was predicted by ARI in Zone 2 (XZ and its adjacent areas)
and Zone 3 (SY and its adjacent areas), which, however, never occurred in CTL experiment.
Concerning temporal variations, 3-hour precipitation rates for these three zones derived from
TRMM 3B42 retrievals are plotted in Figure 9. Compared to CTL, ARI experiment succeeded
in capturing the approximate onset time for all the three regions.





The backward trajectories in Figure 2a, which were calculated by HYSPLIT (Hybrid Single
Particle Lagrangian Integrated Trajectory Model, Draxler and Rolph, 2003), clearly indicate
that the air masses had passed through fire-dense regions before approaching NJ, XZ and SY.
Then cloud evaporation in Zone 1 was predicted by ARI, and meanwhile Zone 2 and 3
experienced increased nocturnal rainfall on 10 June in ARI experiment. In other words, the
spatiotemporal shifts in precipitation during this biomass burning event were closely
connected with aerosols' radiative effects.
3.3.1   Suppressed daytime precipitation
Over Zone 1, CTL produced a convective rainfall event in the afternoon that actually did not
happen, while ARI simulation with no precipitation was closer to the observations. From
energy budget and radiation flux calculation (Figure 5), on 10 June more than 6 MJ/m$^2$ solar
radiation that supposed to reach the surface was blocked in the atmosphere over Zone 1. The
presence of light-absorbing aerosols reduced sensible heat flux and evapotranspiration at the
surface (Figure 6). Large-eddy simulation for biomass burning regions of Brazil deduced that
the peak reductions in sensible and latent heat flux were 60 and 70 W/m$^2$ (Feingold et al.,
2005), which are quantitatively similar to those near NJ estimated in this work. It was shown
that reduced surface flux alone was sufficient to explain the observed cloud dissipation during
the biomass burning event in Brazil. For this case, this convective rain got disappeared merely
by nudging 2-m temperature in the WRF modelling run by Ding et al. (2013a), highlighting
the importance of surface flux modification in the development of these convective clouds.
To figure out the role of vertical thermal behaviors in Zone 1, temporal variations of zone-
averaged differences in temperature, relative humidity (RH) profiles between ARI and CTL
runs are illustrated in Figure 10a and b. From 9:00 LT in the morning, a 1-km-thick belt with
BC-laden smoke approached Zone 1 and covered over the boundary layer top. The radiative
extinction by the elevated smoke layer led to a cooling effect at the surface, which reduced the
boundary layer height and decreased the air temperature in the boundary layer.
Simultaneously, relatively strong warming effect between the altitudes of 1-3 km increased
the air temperature above the boundary layer. The cooling at the lower altitude and warming
at the upper altitude made the stability significantly increased, especially near the top of the
boundary layer, which further suppressed the development of boundary layer. For the
humidity perturbations, the enhanced stability reduced the boundary layer height and hindered
the upward transport of water vapor to a higher altitude, while the heating aloft decreased RH





by increasing the air temperature there. These led to a resultant decrease of more than 20% in
RH above the boundary layer. A more stable and shallower boundary layer in ARI tended to
reduce convective mixing and effectively cut off the cloud layer from its source of moisture,
subsequently desiccating the cloud layer, and leading to substantially weakened vertical
motions. Accordingly, ARI-simulated updraft velocity above 1 km was only one-tenth that of
CTL experiment in the afternoon of 10 June, as demonstrated in Figure 10f. Therefore,
compared with CTL, less water vapor condensed above 1 km but accumulated beneath due to
much weaker convection in ARI experiment (Figure 10e).
In addition to Zone 1, this warmed belt was also blanketing a wider range from 116 to 120 °E
at the moment when the CTL-predicted rainfall started (Figure 9a shows that the rainfall
occurred around 14:00 LT), as shown in the longitude-height cross sections of temperature
difference between CTL and ARI experiment in Figure 10c. In CTL run, cumulus cloud layer
appeared above the inversion capping the boundary layer (Figure 10d). However, the
absorbing aerosol in ARI run heated the atmosphere aloft and stabilized the sub-cloud layer.
The decrease in specific humidity was collocated with warmed upper air since that
atmospheric heating and surface cooling weakened vertical convection and further reduced
the vertical transport of water vapor. Lower entrainment rate together with higher saturation
pressure resulted in daytime decoupling and thinning of the cloud layer all along the longitude
from 116 to 120 °E. This effect might be further strengthened by a positive feedback loop as
described by Jacobson (2002) in which cloud loss leads to an increasing opportunity for BC's
light absorption.
3.3.2   Enhanced nocturnal precipitation
A precipitation rate of over 2.5 mm/h was observed around 19:00-20:00 LT on 10 June in XZ
and its surrounding areas (Zone 2). Only ARI simulation captured this precipitation event. As
shown in Figure 11a, there existed two layers with a high BC concentration of up to10 $\mu g/m^3$
during daytime over Zone 2. One was near the surface and peaked around 18:00 LT, which
could be linked to local fire emissions. The other one was lying over the boundary layer top,
which was apparent at an altitude of 0.8 km before the boundary layer developed and at 2 km
after 15:00 LT. It was very likely to be associated with the transport of upstream fire pollution.
Owning to strong radiative heating effect of BC, a warmer layer was formed above 1 km
during daytime with temperature increase over 1.0 °C. On the contrary, near-surface
temperature kept decreasing. The decline reached its maximum around 20:00 LT. It was also



supported by Figure 7b in which the near-surface temperature decreased by almost 8.0 ℃ at
XZ. Until 16:00 LT, the upper-air warming due to radiative absorption was gradually
compensated by cooling from the surface through vertical mixing. Changes in RH were
almost opposite of those in air temperature. Around 18:00 LT, RH at 3-km altitude started to
increase and then a precipitating cloud formed there.
To get a better insight on the dynamical processes that contribute to precipitation change,
longitude-height cross section of zonal mean responses of temperature, water vapor and wind
profile just before the onset time of precipitation are demonstrated in Figure 11c and d.
Noteworthy is that warmed upper air between 117 to 119 ℉ led to less condensation there.
More water vapor accumulated below 1 km and was then transported toward Zone 2 by the
prevailing east wind near the surface, leading to an excess water vapor over Zone 2 in ARI
(Figure 11e). Simultaneously, radiatively heated air parcel with a temperature increase of 0.5
℃ was found around 2 km over Zone 2. The warmer layer around 2-3 km combined with
large drops in temperature beneath resulted in a buoyancy-driven lifting force. Moreover,
horizontal heterogeneity in atmospheric heating provided the low-level convergence for
maintaining convection in a conditionally unstable atmosphere around 3 km. Thus the zone-
averaged updraft velocity in ARI tripled that predicted by CTL at the altitude of 3 km when
the precipitation began (Figure 11f). Understandably, what made the precipitating cloud
formed around 3 km over Zone 2 were the accumulated moisture near the surface and
anomalous updraft of the air that favored the vertical uplift of water vapor. The release of
latent heat may increase the upper-air instability and further enhance the precipitation.
For the downwind region Zone 3, the warming effect caused by aerosol-radiation interaction
was evident for the air column above 0.5 km all day long on 10 June (Figure 12a). The
warming pattern was coincident with the distribution of BC concentration since BC is the
predominant light-absorbing aerosol specie in the atmosphere. As a result of increased air
temperature, RH decreased substantially during daytime. At late night, an extra precipitating
cloud formed above 2 km over Zone 3 in ARI simulation, leading to a nocturnal precipitation
with a strength of approximately 6 mm/h at 01:00 LT on 11 June. What triggered this rainfall
event is a bit complicated than that over Zone 2. First, the whole air column was getting
cooled at the moment when the precipitation took place, inevitably raising RH value. The RH
increase was quite apparent at the altitude of 3-4 km. Second, daytime radiative absorption by
BC-laden plumes around 2 km heated the surrounding air. Relatively warmer layer at the



altitude of ~ 2 km generated a positive buoyant updraft (Figure 12f), hence air parcel there
was displaced upwards along with enhanced convergence carrying in moist air. This effect
has been proposed by Fan et al. (2015) as part of termed "enhanced conditional instability",
by which absorbing aerosols escalate convection downwind of a heavily polluted area and
promote precipitation. Last but not the least, spatially heterogeneous aerosol-related heating
was associated with greater horizontal temperature lapse, resulting in a convergence flow
above 3 km with an additional onshore wind (Figure 12d). Zone 3 is only about 20 km from
the Yellow sea. It is plausible that more water vapor-saturated air masses originating from the
ocean brought in excess water vapor and consequently elevated the humidity above 3 km
(Figure 12e). We suggest that these precipitating clouds formed because of instability at the
top of the smoke layer, driven by the strong radiation absorption that warmed the surrounding
air. Therefore, the heated BC-laden air was ascended and cooled, leading to the formation of
clouds preferentially in the conditionally unstable zone in the upper air.
**3.4    Uncertainties**
Though the modelling work here characterized cloud and precipitation anomalies during the
biomass burning event, we may also question to what extent the modelling reproduced the
relevant processes in the real world. As widely acknowledged, accurate simulation of smoke
plume and prediction of clouds are both challenging for regional/global models. One
contributor to the uncertainties is the characterization of fire emission. The magnitude was
determined by statistical information and laboratory experiment data, whose accuracy and
representativeness may introduce some uncertainties. The spatiotemporal distribution of fire
emission was allocated based on MODIS retrievals. Loss of information due to cloud
coverage and poor detection efficiency of short-lived or small-scale fires are major limitations
(Giglio et al., 2003). Another challenge is quantification of heat release from biomass burning
and subsequent effects on local and regional meteorology. Furthermore, much emphasis has
been paid to the vertical distribution of absorbing aerosol, to which the cloud response is
highly sensitive (Koch and Del Genio, 2010). The vertical profile of absorbing aerosol in this
simulation underwent little constrain due to limited observation at that time. The regional
model is hardly capable of precisely presenting turbulent flows and vertical transport, thus
introducing uncertainties in three dimensional distributions of BC. It also should be noted that
BC is co-emitted with other components such as OC and sulfur dioxide that oxidizes to
sulfate (Xie et al., 2015). Mixing with other scattering aerosol would considerably amplify the



absorbing efficiency of BC. Model's ability to account for the evolution of mixing state and
how to quantify its amplification also affect the simulated radiative behaviors. Besides, poorly
recognized secondary organic carbon (SOC) formation processes and its light absorption
makes it imperative to reassess and redefine the chemical mechanism and optical properties of
OC in models (Saleh et al., 2014). The large uncertainty in simulating clouds and further
aerosol-cloud interaction is another limitation (e.g., Wang et al., 2011; Tao et al., 2012). To
improve the model performance in all these chemical and physical processes, more
comprehensive measurements and modelling efforts are needed in the future.

**4    Conclusions**
To investigate radiative effects of aerosol-radiation interaction on cloud and precipitation
modifications during the exceptionally active agricultural fire season in June 2012, a bottom-
up emission inventory of crop open burning was developed and then the fully coupled online
WRF-Chem model was applied in this work. The evaluation of simulation through ground-
based observations and satellite retrievals showed that the model generally captured spatial
patterns and temporal variations of fire pollution, which was predominantly concentrating
over northern Anhui and central-north Jiangsu. It is evident that post-harvest burning of crop
residues emitted a tremendous amount of atmospheric pollutants and deteriorated regional air
quality in East China. Elevated concentration of aerosols, particularly light-absorbing BC,
would heat the atmosphere and cool the ground surface through both direct solar radiation
attenuation (direct radiative forcing) and cloud redistribution (semi-direct radiative forcing).
This radiative cooling (heating) effects were distinct close to (downwind from) the source
regions of fire sites. Adjusted temperature structure was intimately linked to small-scale
processes such as turbulent mixing, entrainment and the evolution of the boundary layer.
Subsequently, over NJ and its adjacent regions, absorbing aerosols immediately above the
boundary layer top increased the inversion beneath, reducing available moisture and leading
to a burn-off effect of cloud. Meanwhile, fire plumes played an enhancement role in nocturnal
precipitation over northern Jiangsu by increasing up-level convective activity and fostering
low-level convergence that carries in more moist air. Overall, aerosols' radiative effect on
precipitation modification is therefore likely to depend to a large extent on local
meteorological conditions like atmospheric instability and humidity.



**Acknowledgements**
This work was supported by the National Natural Science Foundation of China
(D0512/41422504, D0512/91544231, and D0510/41505109). Part of this work was supported
by the Jiangsu Provincial Science Fund for Distinguished Young Scholars awarded to A. J.
Ding (No. BK20140021).






Table 1. WRF-Chem modelling configuration options and settings.

| Domain setting | | |
|---|---|---|
| | Domain 1 | Domain 2 |
| Horizontal grid | 130×130 | 160×160 |
| Grid spacing | 20 km | 4 km |
| Vertical layers | 31 | 31 |
| Configuration options | | |
| Long-wave radiation | RRTMG | |
| Short-wave radiation | RRTMG | |
| Land-surface | Noah | |
| Boundary layer | YSU | |
| Microphysics | Lin et al. | |
| Cumulus parameterization | Grell–Deveny (only for domain 1) | |
| Photolysis | Fast-J | |
| Gas-phase chemistry | CBMZ | |
| Aerosol scheme | MOSAIC | |







Table 2. Statistical analyses of the simulated 2-m temperature and the corresponding
observations at four different cities.

|  | MB[a] | | ME[a] | | RMSE[a] | |
|---|---|---|---|---|---|---|
|  | CTL | ARI | CTL | ARI | CTL | ARI |
| NJ | 0.85 | 0.37 | 1.70 | 1.66 | 2.39 | 2.15 |
| BB | 2.19 | 0.98 | 2.51 | 1.65 | 3.27 | 2.16 |
| XZ | 1.67 | 0.51 | 2.37 | 2.19 | 3.32 | 2.89 |
| SY | -0.28 | -0.46 | 1.97 | 1.65 | 2.52 | 2.03 |

[a]MB, ME and RMSE refer to mean bias, mean error and root-mean-square error respectively.





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



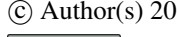

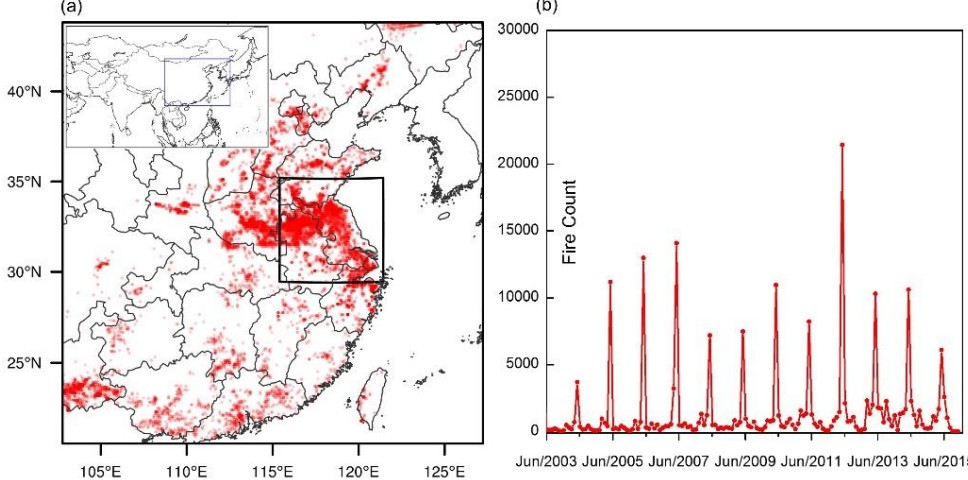

**Figure 1. (a)** Distribution of 13-year fire detections by MOD14A1 during 2003–2015 in the model domain. The black rectangle represents the inner domain. The top left corner gives a map showing the geographic location of the model domain. **(b)** 13-year time series of monthly fire detections in the model domain based on MOD14A1 retrievals.



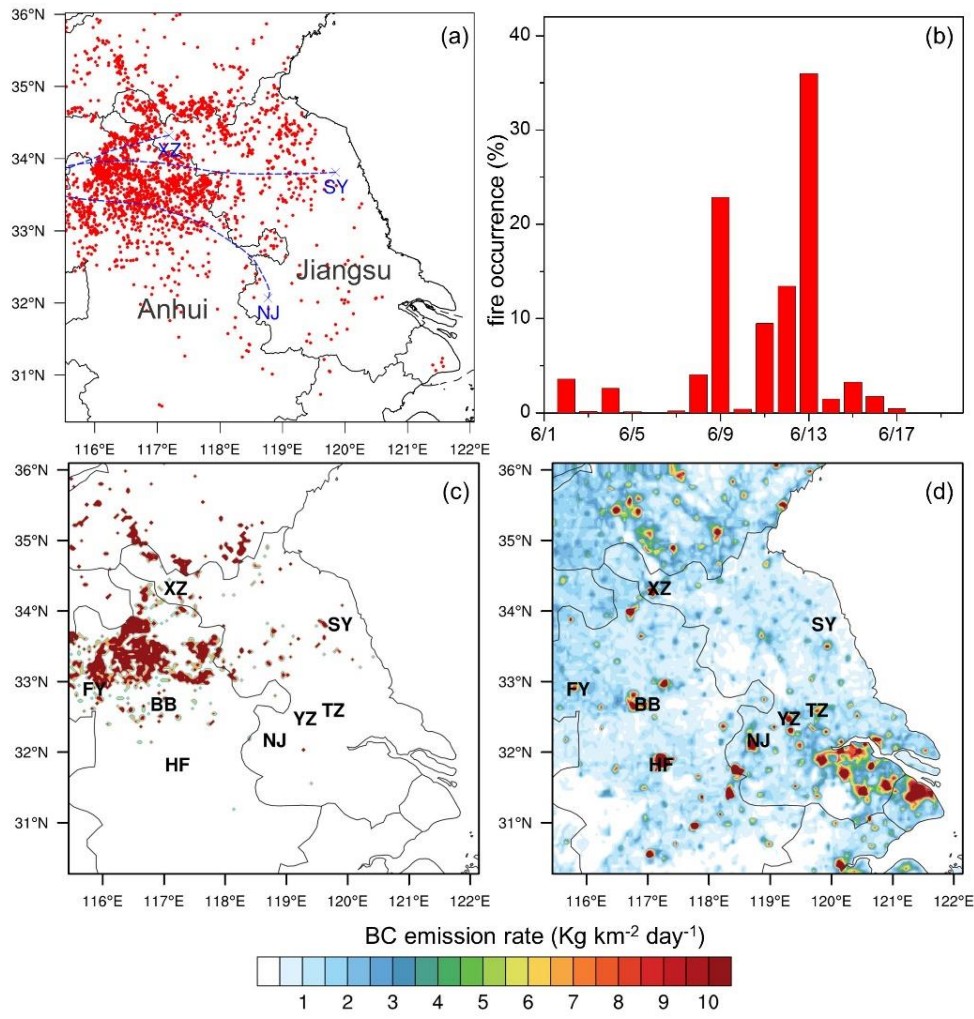

761

**Figure 2. (a)** Satellite fire detections in June 2012 and backward trajectories for NJ (Nanjing),

XZ (Xuzhou) and SY (Sheyang), **(b)** Temporal variations of daily fire occurrences. BC

emission rates from **(c)** agricultural fires and **(d)** anthropogenic activities on 9 June.  Note:

The backward trajectories in (a) was calculated for an altitude of 2 km over NJ, XZ, and SY

from 14:00 LT, 18:00 LT on 10 June and 01:00 on 11 June.





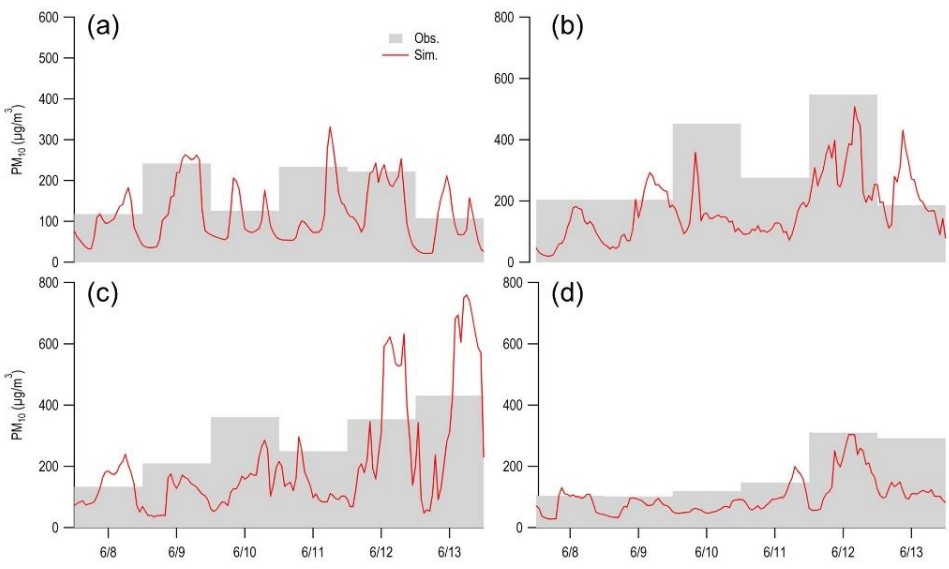


**Figure 3.** Measurements of 24-hour averaged PM10 concentrations and hourly PM10
simulations at **(a)** FY (Fuyang), **(b)** BB (Bengbu), **(c)** XZ (Xuzhou) and **(d)** HF (Hefei).



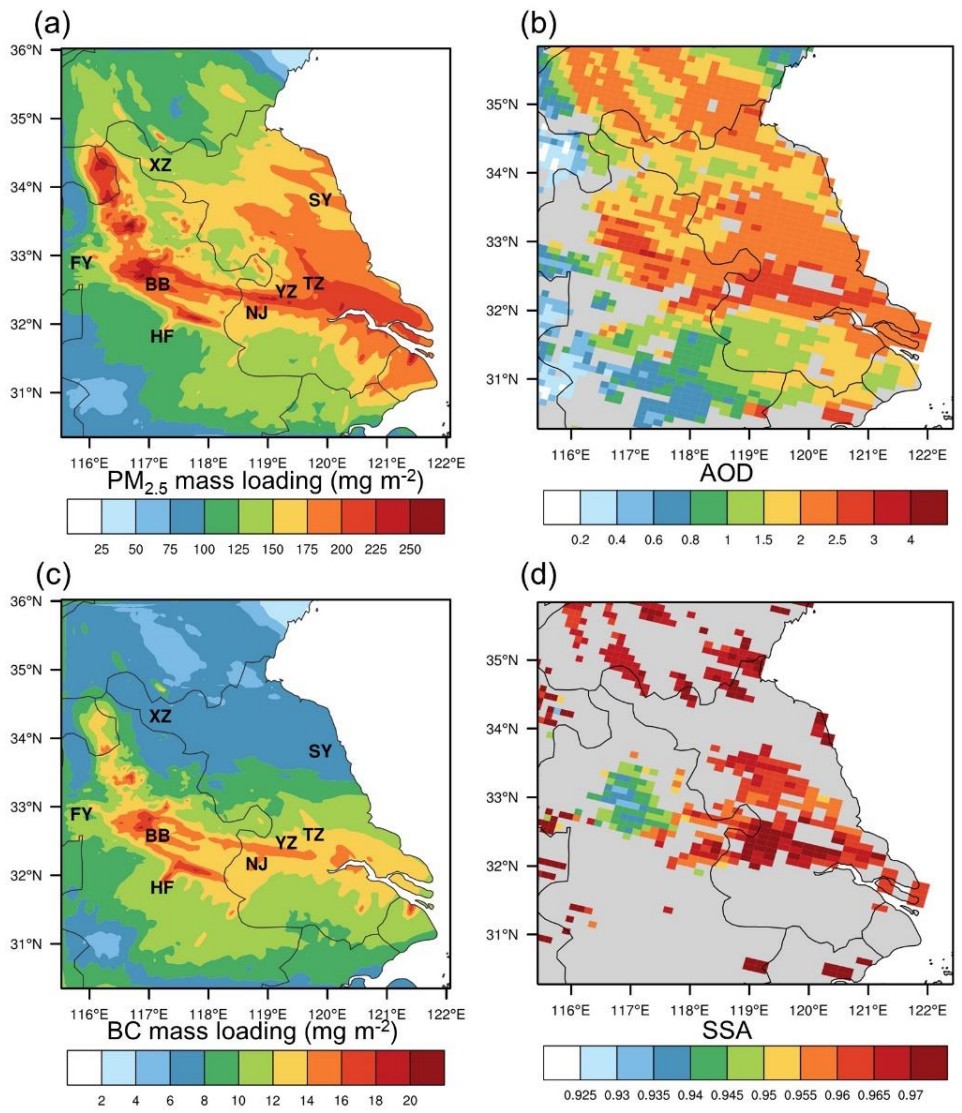


**Figure 4.** Spatial distributions of **(a)** simulated PM2.5 mass loading and **(b)** satellite-derived
660-nm AOD at 11:00 LT, 9 June. **(c)** simulated BC mass loading and **(d)** satellite-derived
SSA at that time.







**Figure 5.** Radiative forcing of aerosol **(a)** at the surface and **(b)** in the atmosphere on 10 June. Spatial pattern of daily averaged BC mass concentrations **(c)** near the surface and **(d)** at the altitude of 2 km. Aerosol-induced changes in air temperature and wind fields **(e)** near the surface and **(f)** at the altitude of 2 km.





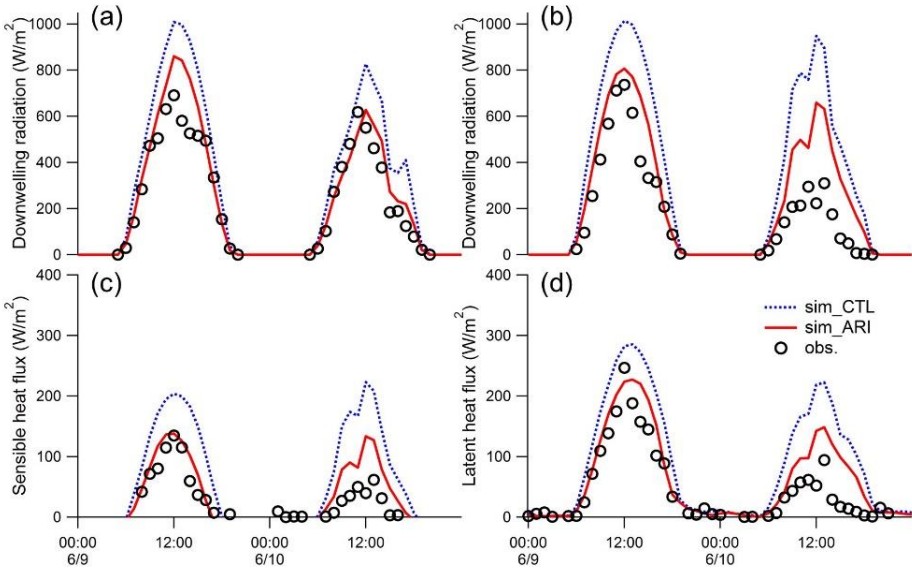


**Figure 6.** Diurnal variations of simulated and observed downwelling short-wave radiation at
**(a)** HF (Hefei) and **(b)** NJ (Nanjing) on 9-10, June. Comparisons of simulated sensible **(c)** and
latent heat fluxes **(d)** with the measurements at NJ. Blue and red lines mean CTL and ARI
simulation. Black circles mark the observations.





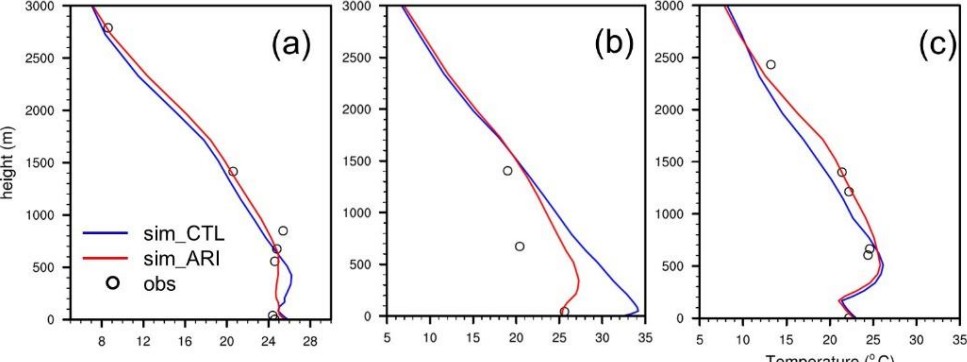


**Figure 7.** Comparisons between the observed and modelled air temperature profiles for **(a)** NJ
(Nanjing) at 08:00LT, **(b)** XZ (Xuzhou) and **(c)** SY (Sheyang) at 20:00 LT, 10 June. Black
circles denote sounding observations. Blue and red solid lines are numerical experiments
without (CTL) and with radiative effects of aerosols (ARI), respectively.





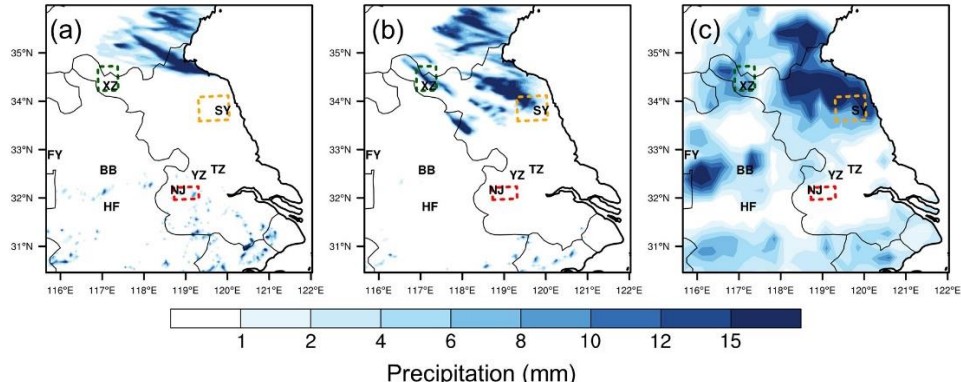

789

**Figure 8.** Modelled precipitation during the period from 00:00 UTC, 10 June to 00:00 UTC, 11 June while excluding and considering radiative effects of aerosols in CTL **(a)** and ARI **(b)** experiment. **(c)** Corresponding TRMM-observed precipitation. Three regions with notable changes in precipitation are marked in rectangles: Zone 1 (red dashed line), Zone 2 (green dashed line) and Zone 3 (yellow dashed line).





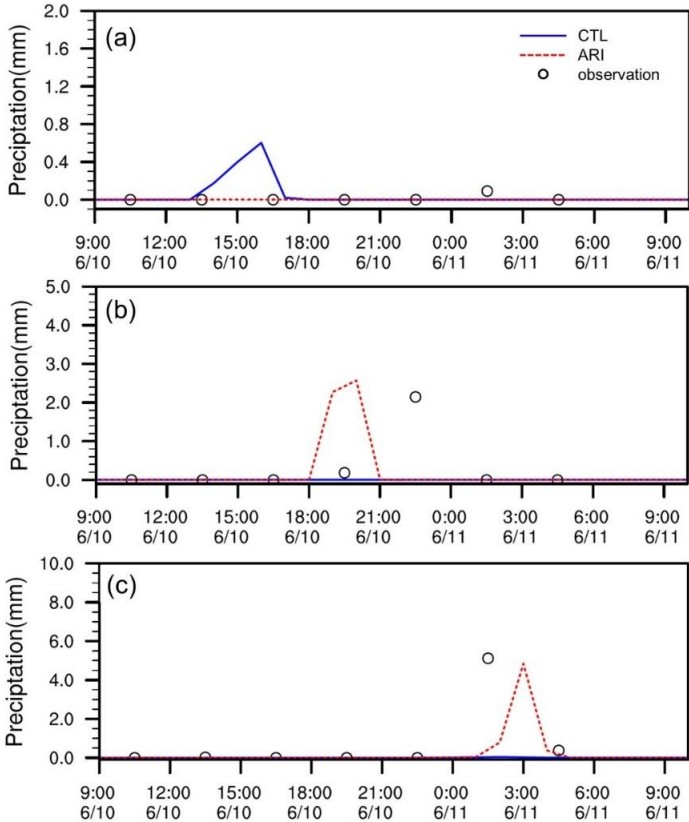

795

**Figure 9.** Simulated hourly precipitation while considering (red dashed lines, ARI) and excluding (blue solid lines, CTL) radiative effects of aerosols, and their comparisons with TRMM observations (black circles) for Zone1 **(a)**, Zone 2 **(b)** and Zone 3 **(c)**.



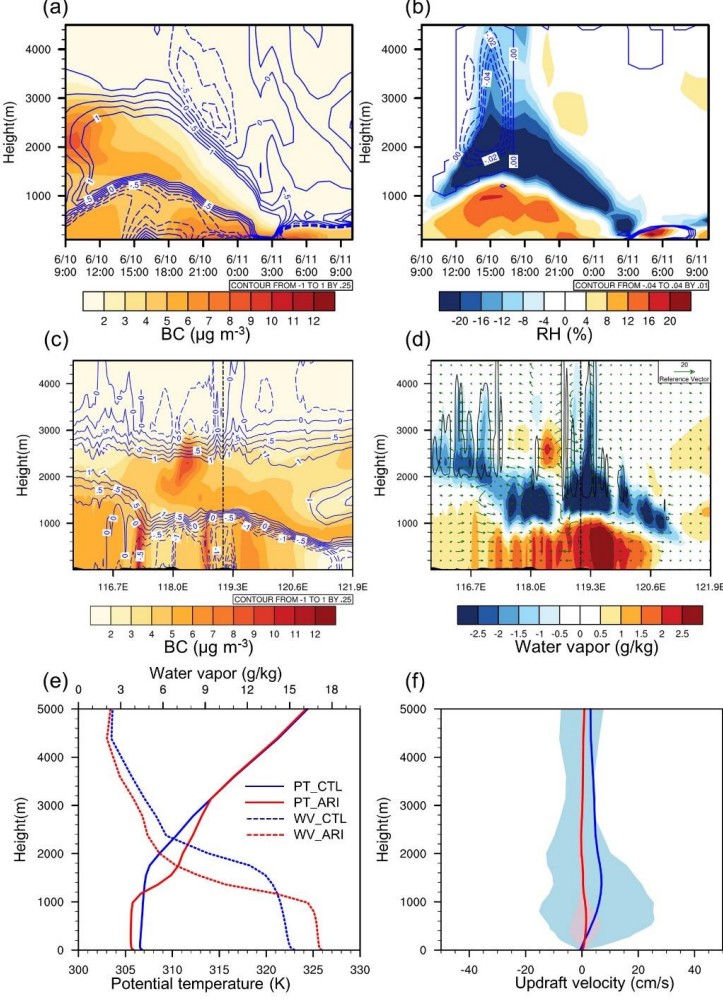

799

**Figure 10. (a)** Temporal evolutions of BC vertical profile and changes in air temperature (K),
**(b)** perturbations in RH (%) and cloud water (g kg1) over Zone 1. **(c)** Longitude-height cross
sections of BC concentrations and aerosol-induced temperature changes at 14:00 LT, 10 June.
**(d)** same as (c) but for water vapor (g kg$^{-1}$) and wind fields (m s$^{-1}$). Note that the vertical wind
speed was multiplied by a factor of 100. Red and black lines in (d) outline cloud coverage
(cloud water mass ratio greater than $10^{-3}$ g kg$^{-1}$) in ARI and CTL simulation. In this case, the
condensate mass ratio was less than $10^{-3}$ g kg$^{-1}$ for the whole column in ARI, thus no red line
is presented in d. **(e)** Vertical profile of zone-averaged potential temperature (PT) and water
vapor ratio (WV), and **(f)** updraft velocity predicted by ARI (red) and CTL (blue) at 14:00 LT.
Shadows in f represent 25-75 percentile range of simulated updraft velocity.





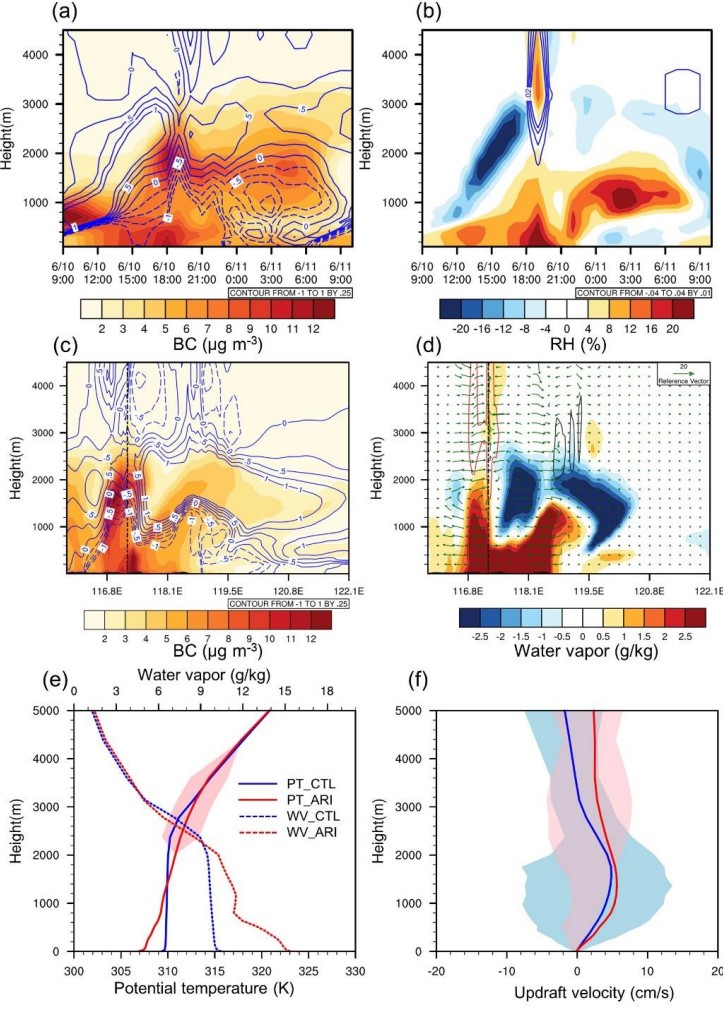

810

**Figure 11. (a)** Temporal evolutions of BC vertical profile and changes in air temperature (K),
**(b)** perturbations in RH (%) and cloud water (g kg1) over Zone 2. **(c)** Longitude-height cross
sections of BC concentrations and aerosol-induced temperature changes at 18:00 LT, 10 June.
**(d)** same as (c) but for water vapor (g kg$^{-1}$) and wind fields (m s$^{-1}$). Note that the vertical wind
speed was multiplied by a factor of 100. Red and black lines in (d) outline cloud coverage
(cloud water mass ratio greater than $10^{-3}$ g kg$^{-1}$) in ARI and CTL simulation. **(e)** Vertical
profile of zone-averaged potential temperature (PT) and water vapor ratio (WV), and **(f)**
updraft velocity predicted by ARI (red) and CTL (blue) at 18:00 LT. Shadow in e marks
conditionally unstable zone in the upper air in ARI. Shadows in f represent 25-75 percentile
range of simulated updraft velocity.

821





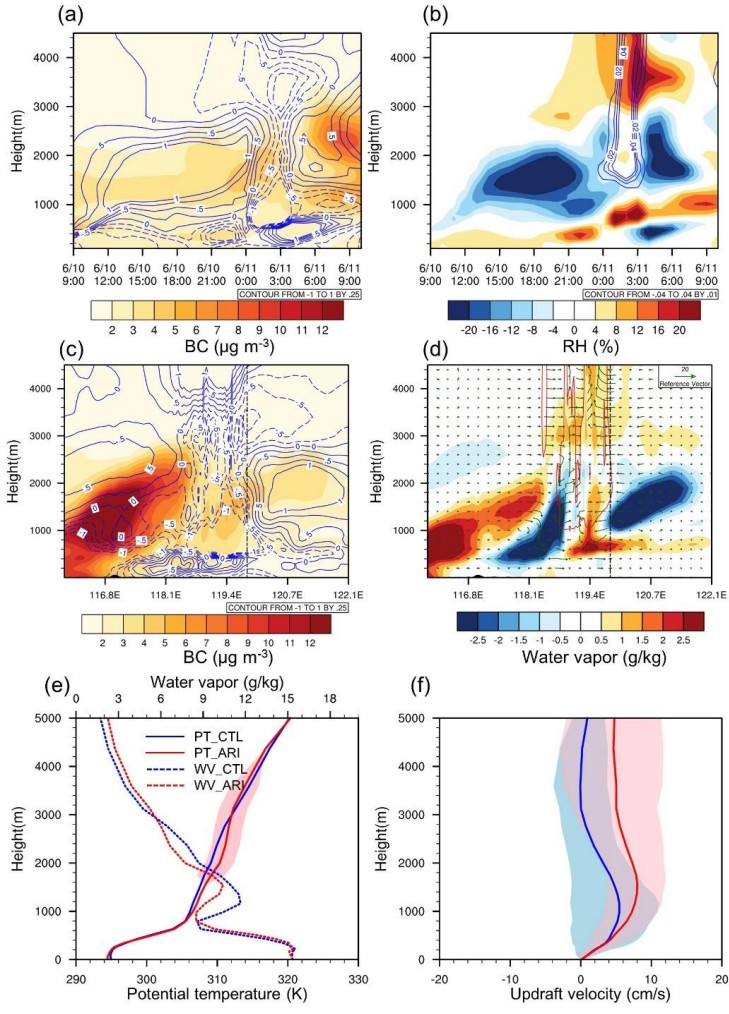

822

Figure 12. (a) Temporal evolutions of BC vertical profile and changes in air temperature (K),
(b) perturbations in RH (%) and cloud water (g kg1) over Zone 3. Longitude-height cross
sections of BC concentrations and aerosol-induced temperature changes at 01:00 LT, 11 June
(c). (d) same as (c) but for water vapor (g kg$^{-1}$) and wind fields (m s$^{-1}$). Note that the vertical
wind speed was multiplied by a factor of 100. Red and black lines in (d) outline cloud
coverage (cloud water mass ratio greater than 10$^{-3}$ g kg$^{-1}$) in ARI and CTL simulation. In this
case, the condensate mass ratio was less than 10$^{-3}$ g kg$^{-1}$ for the whole column in CTL, thus no
black line is presented in d. (e) Vertical profile of zone-averaged potential temperature (PT)
and water vapor ratio (WV), and (f) updraft velocity predicted by ARI (red) and CTL (blue) at
01:00 LT. Shadow in e marks conditionally unstable zone in the upper air in ARI. Shadows in
(f) represent 25-75 percentile range of simulated updraft velocity.