# Peer review of "Effects of aerosol-radiation interaction on precipitation during"

_Atmospheric Chemistry and Physics, 2016_

## Short Comment (SC1) · 8 Apr 2016

Very nice, meaningful and thoughtful work on weather modification by aerosols. I have two small suggestions.

1. It would be interesting to compare meteorology changes induced only by biomass burning with changes induced by aerosols from anthropogenic emissions, since the aerosol concentrations in Nanjing are still high without fire.

2. The changes in meteorology highly depend on concentrations of BC. The results would be more convincing if BC is verified using aerosol composition measurements or AAOD (aerosol absorption optical depth) data.

[Figure]

I also studied aerosol-radiation interaction during high aerosol loading event. You can check it if you have interest.

http://www.atmos-chem-phys.net/16/1673/2016/

Thanks for the nice work and contribution to the scientific community.

---

## Referee Comment (RC1) · Anonymous Referee #1 · 21 May 2016

General Comments: This is a very interesting study to investigate the impacts of agriculture fire emissions on temperature, precipitation, and clouds over East China. The study selected a typical event around June 10 2012 and conducted model simulations using WRF-Chem to examine the impacts. The results show that the absorbing aerosols emitted from the agriculture fire interacted with radiation and changed the meteorological conditions. This redistributes the precipitation over the downwind areas of the burning plumes. The results are well presented, and the topic is suitable for publication in ACP after addressing some specific comments listed below.

Specific Comments: 1. Since this study investigated the impacts of fire emissions on meteorological fields, more discussion about the fire emission inventory may be

[Figure]

needed. In Section 2.1, please state what's the spatial and temporal resolution of the fire emission inventory used in this study. Section 3.4 discussed about the uncertainties that are partly from the fire emission spatial and vertical distributions. Did you compare your emission inventory with the FINN fire emission data (Wiedinmyer et al., 2011) that are with hourly temporal and 1x1 km2 horizontal resolutions? In terms of vertical distribution of fire emissions, did you use the plumerise scheme in WRF-Chem or prescribe distribution profile? Please clarify.

Wiedinmyer, C., Akagi, S. K., Yokelson, R. J., Emmons, L. K., Al-Saadi, J. A., Orlando, J. J., and Soja, A. J.: The Fire INventory from NCAR (FINN): a high resolution global model to estimate the emissions from open burning, Geosci. Model Dev., 4, 625-641, doi:10.5194/gmd-4-625-2011, 2011.

2. In Section 2.2, the simulation was conducted from May 20 to June 15, but the analysis was only for June 9-11. This is confused. I would suggest just mentioning that you have a simulation period for spin-up the chemistry initial condition. More importantly, please state what's the meteorological initializing date for the event of June 9-11. The different initializing date may change the results of impacts. Did you try different initializing date to see whether the results changed?

3. In line 175 of page 6, I am not convinced that the ACI should be disabled for investigating ARI effect. Please explain why ACI should be disabled in this study? I think it will be more interesting to compare both ACI and ARI. Although authors pointed that previous studies found ARI sometimes is more important, this is not always the case (e.g., Zhong et al., 2015). Another critical issue of turning off ACI in WRF-Chem is about aerosol wet removal. In WRF-Chem, aerosol wet removal is linked with ACI. With turning off ACI, please clarify how you treat the wet removal of aerosols in your simulations since your event (Fig. 8) shows significant amounts of precipitation.

Zhong, S., Y. Qian, C. Zhao, R. Leung, and X.-Q. Yang (2015), A case study of urbanization impact on summer precipita-tion in the Greater Beijing MetropolitanArea: Urban heat island versus aerosoleffects, J. Geophys. Res. Atmos., 120,10,903–10,914, doi:10.1002/2015JD023753.

4. This study highlighted the impacts of fire emissions, however, the experiments were only designed with CTL and ARI. Based on these two experiments, it's hard to disentangle the biomass burning effect from the effects of anthropogenic aerosols. One experiment without biomass burning emissions is needed if the purpose is to investigate the impacts of agriculture fire.

5. In lines 219-221 of page 8, please provide the corresponding simulated values as well.

6. In Fig. 3, do you have hourly observations? If not, why not also put daily mean of simulated values for a direct comparison?

7. In Fig. 4, why not show the simulated SSA?

8. Section 3.3, Fig. 8, it seems to me that model has large biases in capturing spatial distributions of TRMM precipitation. Can you try another precipitation dataset (CMORPH) for comparison? CMORPH provides 8 km resolution data. Is this poor comparison between model and TRMM due to the initial condition? Did you try different initial meteorological conditions? In addition, this may be also partly due to the missing of aerosol-cloud interaction? Strong suggestion to test this case with aerosol-cloud interaction.

9. I don't see the necessity to include the paragraph of Line 331-337 of page 12.

---

## Referee Comment (RC2) · Anonymous Referee #2 · 1 Jun 2016

The authors provides an interesting and instructive work on the radiation effect of aerosols from biomass burning and the impacts on meteorological parameters including clouds, temperature, relative humidity, and rainfall in East China by conducting two parallel numerical simulations with the online coupled model WRF-Chem. The experiments are well designed and the presented results are generally convinced. Overall, I believe that it is a valuable study to highlight the importance of straw burning in weather modifications as well as air quality deterioration. It is worth to be published in ACP after adding more in-depth discussion of the simulated results. The specific comments are shown as follows.

Main issues:

[Figure]

As pointed out in the present work, BC was the most important factor that alters the radiation budget. However, in the ARI simulation, BC was emitted not only from crop straw burning, but also from residential combustion and transportation, especially in anthropogenic emission-intensive region like East China. It is hard to figure out the effects of straw burning through these two simulations. Thus, this work would be further improved by isolating the radiative forcing just caused by straw fires during this biomass burning case. Quantitative comparisons between radiative effects induced by agricultural fires and anthropogenic pollutants could make more sense.

Some of the detailed descriptions on the method part need to be clarified, for instance, how the estimated emissions were allocated using MODIS detections. It should be noted that detected fire spots could be caused by forest or grassland fires, rather than crop straw burning.

Another deficiency of this paper is that while discussing the precipitation redistribution in Section 3.3, there is a lack of an in-depth analysis of how or through which processes the fire plumes influence the temperature stratification and moisture conditions. It is can be further improved by adding some diagnoses like CAPE (convective available potential energy) or MSE (moist static energy), which may provide more direct evidence (Fan et al., 2015).

In addition to the main concerns above, additional minor comments are given below.

Minor comments:

1) Line 173 and 174: Define ARI and CCN when they first appeared. 2) Line 214: Change Air Pollution Index (API) to Air Quality Index (AQI) 3) Caption of Fig.10: the unit of cloud water mixing ratio should be "g kg-1". 4) Line 360: the sentence "For the humidity perturbations" is better to be rephrased to "the perturbations in humidity".

Reference: Fan, J. W., Rosenfeld, D., Yang, Y., Zhao, C., Leung, L. R., and Li, Z. Q.: Substantial contribution of anthropogenic air pollution to catastrophic floods in Southwest China, Geophys. Res. Lett., 42, 6066-6075, 10.1002/2015GL064479, 2015.

---

## Author Comment (AC1) · 7 Jul 2016

**Response to Short Comment #1**

*Very nice, meaningful and thoughtful work on weather modification by aerosols. I have two small suggestions.*

*1. It would be interesting to compare meteorology changes induced only by biomass burning with changes induced by aerosols from anthropogenic emissions, since the aerosol concentrations in Nanjing are still high without fire.*

**Response:** Thanks a lot for the suggestions, which helps improve this work. Indeed, we agree that comparing the effect of biomass burning and anthropogenic pollutions would make it more clear. Thus, in the revision, we will add another experiment that only includes anthropogenic emissions to better understand the role of agricultural fires in the weather modification.

*2. The changes in meteorology highly depend on concentrations of BC. The results would be more convincing if BC is verified using aerosol composition measurements or AAOD (aerosol absorption optical depth) data.*

**Response:** Indeed, it would be better to evaluate the simulation using AAOD retrievals. However, there are too much invalid data during that period. That why we did not apply AAOD retrievals in the article.

[Figure]

Fig. R1 OMI retrievals of aerosol absorption optical depth at 388 nm on 9 and 10 June 2012.

---

## Author Comment (AC2) · 7 Jul 2016

**Response to Referee #1**

*General Comments: This is a very interesting study to investigate the impacts of agriculture fire emissions on temperature, precipitation, and clouds over East China. The study selected a typical event around June 10 2012 and conducted model simulations using WRF-Chem to examine the impacts. The results show that the absorbing aerosols emitted from the agriculture fire interacted with radiation and changed the meteorological conditions. This redistributes the precipitation over the downwind areas of the burning plumes. The results are well presented, and the topic is suitable for publication in ACP after addressing some specific comments listed below.*

**Response:** We would like to thank the referee for providing the insightful suggestions, which indeed help us further improve the manuscript.

*Specific Comments: 1. Since this study investigated the impacts of fire emissions on meteorological fields, more discussion about the fire emission inventory may be needed. In Section 2.1, please state what's the spatial and temporal resolution of the fire emission inventory used in this study. Section 3.4 discussed about the uncertainties that are partly from the fire emission spatial and vertical distributions. Did you compare your emission inventory with the FINN fire emission data (Wiedinmyer et al.,2011) that are with hourly temporal and 1x1 km2 horizontal resolutions? In terms of vertical distribution of fire emissions, did you use the plume rise scheme in WRF-Chem or prescribe distribution profile? Please clarify.*

*Wiedinmyer, C., Akagi, S. K., Yokelson, R. J., Emmons, L. K., Al-Saadi, J. A., Orlando,J. J., and Soja, A. J.: The Fire INventory from NCAR (FINN): a high resolution global model to estimate the emissions from open burning, Geosci. Model Dev., 4, 625-641,doi:10.5194/gmd-4-625-2011, 2011.*

**Response:** More descriptions on the fire emission inventory will be added in the revised manuscript. We compared the spatial patterns of FINN fire emission data (version 1.5) with the emission inventory used in this work. As shown in Fig. R1, they generally consistent with each other due to the fact that the location and timing for the fires in both inventories are based on MODIS Thermal Anomalies Product. Some inconsistencies like the density of fire in central Jiangsu are attributed to the different land cover dataset applied for identification of underlying biomass type. FINN used MODIS Collection 5 Land Cover Type data while we employed Global Land Cover data. In terms of the spatiotemporal resolution, both the inventories were allocated to daily emissions for each $1 \times 1$ km grid. Technically, time information in FINN dataset shows time of satellite

overpass/observation, not the duration of fire. Our inventory differs from FINN in magnitude. Taking CO emission during the first half of June for the inner model domain, we estimate that 4.5 Tg CO was emitted while FINN gives the value of 7.5 Tg. That is caused by various methods to estimate burned biomass. FINN used MODIS Vegetation Continuous Fields to assign the burned mass. All farmland regardless what kinds of crop are cultivated was assumed to have a fuel loading of 0.5 $kg/m^2$. However, in China, crop straw is used in multiple ways, like biofuel, biogas production and animal feed supply, which is highly dependent on crop species. We estimated the emission using a "bottom-up" method. By fully considering different kinds of crop straw, crop-specific usage and combustion efficiency. The smaller estimation was expectable.

We did not use the plume rise scheme for this biomass burning case, during which the burned biomass is winter wheat straw. Post-harvest crop residue is burned by flaming in mechanized agricultural systems. In contrast, when crops are harvested by hand the residue is often burned in piles that may smolder in China. It is noteworthy that the plume rise scheme is more suitable for the flaming phase (Freitas et al., 2007; Grell et al., 2011). In East China, especially in the northern Jiangsu and Anhui province, most fires of wheat straw are characterized by short-lived, small-scale smoldering (Fig. R2). Correspondingly, the fire radiative power (FRP) between this straw burning is much weaker than the grassland fire in North America (Fig. R3), indicating two different burning conditions. Thus in this work, the straw fire emission was placed in the lowest two levels from the surface to around 50 meter. We will clarify these issues and mention the comparisons of FINN data and emission calculation in the revised manuscripts.

[Figure]

Fig. R1. Comparison of CO emission from biomass burning between FINN V1.5 emission and the emission inventory used in this study during the first half of June 2012. Here, FINN V1.5 dataset was acquired at http://bai.acom.ucar.edu/Data/fire/.

[Figure]

Fig. R2 A photo showing the field burning of wheat straw in Suixi county (33°54′37″N, 116°45′46″E), northern Anhui province on June 14, 2013.

[Figure]

Fig. R3 MODIS-detect maximum fire radiative power (FRP) during crop straw fire in northern Anhui on 9 June 2012 and grassland fire in North America on 3 July 2004.

*2. In Section 2.2, the simulation was conducted from May 20 to June 15, but the analysis as only for June 9-11. This is confused. I would suggest just mentioning that you have a simulation period for spin-up the chemistry initial condition. More importantly, please state what's the meteorological initializing date for the event of June 9-11. The different initializing date may change the results of impacts. Did you try different initializing date to see whether the results changed?*

**Response:** Thanks for the suggestions. The meteorological initializing date was 12:00 UTC on 9 June. We will add the relavant description in the revised manuscript. We have tried different initializing date, there are some discrepancies. Considering higher model performance of first 72-hour forecast and 12-hour spin-up, we think that it is practicable to set 12:00 UTC on 9 June as the initializing date for the simulation of 10 June when

distinct precipitation modifications occurred.

*3. In line 175 of page 6, I am not convinced that the ACI should be disabled for investigating ARI effect. Please explain why ACI should be disabled in this study? I think it will be more interesting to compare both ACI and ARI. Although authors pointed that previous studies found ARI sometimes is more important, this is not always the case (e.g., Zhong et al., 2015). Another critical issue of turning off ACI in WRF-Chem is about aerosol wet removal. In WRF-Chem, aerosol wet removal is linked with ACI. With turning off ACI, please clarify how you treat the wet removal of aerosols in your simulations since your event (Fig. 8) shows significant amounts of precipitation.*

*Zhong, S., Y. Qian, C. Zhao, R. Leung, and X.-Q. Yang (2015), A case study of urbanization impact on summer precipitation in the Greater Beijing Metropolitan Area: Urban heat island versus aerosol effects, J. Geophys. Res. Atmos., 120,10,903–10,914, doi:10.1002/2015JD023753.*

**Response:** This work intended to investigate the effects of aerosol-radiation interaction, as pointed out in the title. Indeed, in some cases, ACI is of great importance, like in Zhong et al. (2015). However, previous studies and review papers indicated that under highly polluted conditions or strongly absorbing aerosol environment, ARI could be the dominant factor (Fan et al., 2008; Rosenfeld et al., 2008; Fan et al., 2015). Sensitivity simulations using cloud parcel model also suggest that CCN activation shows a weaker dependence on aerosol with increasing aerosol loadings, which is converting from an aerosol-limited regime to an updraft-limited regime (Reutter et al., 2009; Chang et al., 2015). We checked the relative sensitivity ratio proposed by Reutter et al.(2009) in Fig. R4. Relative sensitivity ratio less than $0.1\ 10^{-3}\,\mathrm{m\ s^{-1}\ cm^3}$ indicates that CCN activation and cloud drop formation was insensitive to the aerosol concentration during this case.

To further investigate the relative importance of ACI, we conducted another numerical experiment considering both ARI and ACI. Radiative perturbations caused by ARI far outweigh those from ACI at the TOA and at the ground surface, so did the adjustment in vertical temperature profile (Fig. R5-6). ACI cloud be a crucial factor modifying precipitation to large extent under some conditions (Zhong et al., 2015). In Zhong's case, the aerosol loading was much lower. As reported by the ministry of environmental protection of China, $PM_{10}$ concentrations were 126, 62 and 90 $\mu g/m^3$ in Beijing, Tianjin and Shijiazhuang on 28 June 2008. Furthermore, sulfate and other hygroscopic components dominate PM concentration during summer time in Beijing (Zhang et al., 2013), which might enhance aerosols' activity acting as CCN. Besides, stronger updraft motion may further favor aerosol-cloud interaction during the case in

Zhong et al. (2015). By contrast, tremendous freshly-emitted hydrophobic carbonaceous aerosol from biomass burning together with weaker updraft motion may lead to less important role of ACI in this work.

Yes, in WRF-Chem, aerosol wet removal is disabled too if ACI is switched off. In our study, we mainly focused on the initialization of precipitation. As mentioned in the article, the modifications of precipitation onset mainly linked to the vertical re-allocation of short-wave energy due to absorbing aerosol during daytime, which was less affected by wet scavenging later that night. Therefore, ignoring wet scavenging is acceptable here. However, considering the referee's points, we will add a few sentences to address the uncertainty of our treatment.

[Figure]

Fig. R4. Vertical profile of relative sensitivity ratio proposed by Reutter et al., (2009) during the precipitation for three region marked in Fig.8: Zone 1(a), Zone 2 (b), and Zone 3 (c).

[Figure]

Fig. R5. Radiative forcing of ARI at the surface (a) and in the atmosphere (b) on 10 June. Radiative forcing of ACI at the surface (c) and in the atmosphere (d) on 10 June.

[Figure]

Fig. R6. Comparisons between the observed and modelled air temperature profiles for (a) NJ (Nanjing) at 08:00 LT, (b) XZ (Xuzhou) and (c) SY (Sheyang) at 20:00 LT, 10 June. Black circles denote sounding observations. Blue, red and green solid lines are numerical experiments without (CTL), with radiative effects of aerosols (ARI), and with both ARI and ACI (aerosol-cloud interaction), respectively.

*4. This study highlighted the impacts of fire emissions, however, the experiments were only designed with CTL and ARI. Based on these two experiments, it's hard to disentangle the biomass burning effect from the effects of anthropogenic aerosols. One experiment without biomass burning emissions is needed if the purpose is to investigate the impacts of agriculture fire.*

**Response:** Accepted. Another experiment without biomass burning emissions will be conducted and discussed further in the revised manuscript.

*5. In lines 219-221 of page 8, please provide the corresponding simulated values as well.*

**Response:** Accepted. The corresponding simulated values will be added.

*6. In Fig. 3, do you have hourly observations? If not, why not also put daily mean of simulated values for a direct comparison?*

**Response:** Accepted. Fig.3 will be re-plotted using daily mean of simulated $PM_{10}$ concentrations.

*7. In Fig. 4, why not show the simulated SSA?*

**Response:** In the further analysis in section 3.3, BC played a vital role in precipitation modification. Showing its spatial pattern provides a more direct and clear picture of how BC was transported and distributed than that of SSA.

*8. Section 3.3, Fig. 8, it seems to me that model has large biases in capturing spatial distributions of TRMM precipitation. Can you try another precipitation dataset (CMORPH) for comparison? CMORPH provides 8 km resolution data. Is this poor comparison between model and TRMM due to the initial condition? Did you try different initial meteorological conditions? In addition, this may be also partly due to the missing of aerosol-cloud interaction? Strong suggestion to test this case with aerosol cloud interaction.*

**Response:** Fig. R7 shows the comparison of spatial distributions between CMORPH and TRMM satellite-based precipitation on 10 June 2012. As shown, there is no significant difference in spatial patterns. Specifically, no precipitation in Nanjing and the surrounding areas, and precipitation exceeding over 10 mm took place in northern Jiangsu. Indeed, the simulation did not compare well with the satellite observations. We have also tried ECWMF ERA Interim data to initiate the regional model, which cannot well describe the pattern either. As mentioned above, including ACI exert little effect on the radiation flux and temperature stratification. Actually, we have tried a series of simulations with different initial meteorological conditions and various parameterization schemes. Almost all the setups failed to represent such high aerosol loadings (Fig. 3). It is suggested that rapid formation of secondary aerosol, like sulfate and SOA, might be significantly underestimated due to a limited understanding of SOA formation and its optical properties and also heterogeneous chemistry that enhanced the oxidizing capacity of the atmosphere during this biomass-burning case (Xie et al., 2015). Our group will keep working on the secondary transformations of aerosol and the radiative effects during agricultural fires with more in-site measurements.

[Figure]

Fig. R7 (a) Spatial distributions of CMORPH satellite-based daily precipitation on 10 June 2012 (b) Corresponding TRMM-observed precipitation. The CMOPRH data was obtained from http://rda.ucar.edu/datasets/ds502.0/.

*9. I don't see the necessity to include the paragraph of Line 331-337 of page 12.*

**Response:** Accepted. This paragraph will be removed in the revised version.

**References**

Chang, D., Cheng, Y., Reutter, P., Trentmann, J., Burrows, S., Spichtinger, P., Nordmann, S., Andreae, M., Pöschl, U., and Su, H.: Comprehensive mapping and characteristic regimes of aerosol effects on the formation and evolution of pyro-convective clouds, Atmos Chem Phys, 15, 10325-10348, 2015.

Fan, J. W., Zhang, R. Y., Tao, W. K., and Mohr, K. I.: Effects of aerosol optical properties on deep convective clouds and radiative forcing, J Geophys Res-Atmos, 113, Artn D08209 10.1029/2007jd009257, 2008.

Fan, J. W., Rosenfeld, D., Yang, Y., Zhao, C., Leung, L. R., and Li, Z. Q.: Substantial contribution of anthropogenic air pollution to catastrophic floods in Southwest China, Geophys Res Lett, 42, 6066-6075, 10.1002/2015GL064479, 2015.

Freitas, S. R., Longo, K. M., Chatfield, R., Latham, D., Silva Dias, M., Andreae, M., Prins, E., Santos, J., Gielow, R., and Carvalho Jr, J.: Including the sub-grid scale plume rise of vegetation fires in low resolution atmospheric transport models, Atmos Chem Phys, 7, 3385-3398, 2007.

Grell, G., Freitas, S. R., Stuefer, M., and Fast, J.: Inclusion of biomass burning in WRF-Chem: impact of wildfires on weather forecasts, Atmos Chem Phys, 11, 5289-5303, DOI 10.5194/acp-11-5289-2011, 2011.

Reutter, P., Su, H., Trentmann, J., Simmel, M., Rose, D., Gunthe, S., Wernli, H., Andreae, M., and Pöschl, U.: Aerosol-and updraft-limited regimes of cloud droplet formation: influence of particle number, size and hygroscopicity on the activation of cloud condensation nuclei (CCN), Atmos Chem Phys, 9, 7067-7080, 2009.

Rosenfeld, D., Lohmann, U., Raga, G. B., O'Dowd, C. D., Kulmala, M., Fuzzi, S., Reissell, A., and Andreae, M. O.: Flood or drought: How do aerosols affect precipitation?, Science, 321, 1309-1313, 10.1126/science.1160606, 2008.

Xie, Y. N., Ding, A. J., Nie, W., Mao, H. T., Qi, X. M., Huang, X., Xu, Z., Kerminen, V. M., Petaja, T., Chi, X. G., Virkkula, A., Boy, M., Xue, L. K., Guo, J., Sun, J. N., Yang, X. Q., Kulmala, M., and Fu, C. B.: Enhanced sulfate formation by nitrogen dioxide: Implications from in situ observations at the SORPES station, J Geophys Res-Atmos, 120, 12679-12694, 10.1002/2015JD023607, 2015.

Zhang, R., Jing, J., Tao, J., Hsu, S.-C., Wang, G., Cao, J., Lee, C. S. L., Zhu, L., Chen, Z., Zhao, Y., and Shen, Z.: Chemical characterization and source apportionment of PM2.5 in Beijing: seasonal perspective, Atmos. Chem. Phys., 13, 7053-7074, doi:10.5194/acp-13-7053-2013, 2013.

Zhong, S., Qian, Y., Zhao, C., Leung, R., and Yang, X. Q.: A case study of urbanization impact on summer precipitation in the Greater Beijing Metropolitan Area: Urban heat island versus aerosol effects, Journal of Geophysical Research: Atmospheres, 120, 2015.

---

## Author Comment (AC3) · 7 Jul 2016

**Response to Referee #2**

*The authors provides an interesting and instructive work on the radiation effect of aerosols from biomass burning and the impacts on meteorological parameters including clouds, temperature, relative humidity, and rainfall in East China by conducting two parallel numerical simulations with the online coupled model WRF-Chem. The experiments are well designed and the presented results are generally convinced. Overall, I believe that it is a valuable study to highlight the importance of straw burning in weather modifications as well as air quality deterioration. It is worth to be published in ACP after adding more in-depth discussion of the simulated results. The specific comments are shown as follows.*

**Response:** We would like to appreciate the referee for providing such great suggestions. We will conduct more simulations and revise this article.

*As pointed out in the present work, BC was the most important factor that alters the radiation budget. However, in the ARI simulation, BC was emitted not only from crop straw burning, but also from residential combustion and transportation, especially in anthropogenic emission-intensive region like East China. It is hard to figure out the effects of straw burning through these two simulations. Thus, this work would be further improved by isolating the radiative forcing just caused by straw fires during this biomass burning case. Quantitative comparisons between radiative effects induced by agricultural fires and anthropogenic pollutants could make more sense.*

**Response:** Accepted. Another experiment without biomass burning emissions will be conducted and discussed in detail in the revised manuscript.

*Some of the detailed descriptions on the method part need to be clarified, for instance, how the estimated emissions were allocated using MODIS detections. It should be noted that detected fire spots could be caused by forest or grassland fires, rather than crop straw burning.*

**Response:** Accepted. We will add more detailed descriptions on the method of development of the emission inventory and how the crop straw burning was identified.

*Another deficiency of this paper is that while discussing the precipitation redistribution in Section 3.3, there is a lack of an in-depth analysis of how or through which processes the fire plumes influence the temperature stratification and moisture conditions. It is can be further improved by adding some diagnoses like CAPE (convective available*

*potential energy) or MSE (moist static energy), which may provide more direct evidence (Fan et al., 2015).*

**Response:** Accepted. We will add diagnostic parameter MSE to further analyze the modifications in precipitation.

*In addition to the main concerns above, additional minor comments are given below.*

*Minor comments:*

*1) Line 173 and 174: Define ARI and CCN when they first appeared.*

**Response:** Accepted. The definition of the abbreviation will be added.

*2) Line 214: Change Air Pollution Index (API) to Air Quality Index (AQI)*

**Response:** Before the year of 2013, the ministry of environmental protection of China reported API data instead of AQI. Since the study period of this work is 2012, the data we can acquire was API.

*3) Caption of Fig.10: the unit of cloud water mixing ratio should be "g kg-1".*

**Response:** Accepted.

*4) Line 360: the sentence "For the humidity perturbations" is better to be rephrased to "the perturbations in humidity".*

**Response:** Accepted.